# Safety, Immunogenicity and Lot-to-Lot Consistency of Sabin-Strain Inactivated Poliovirus Vaccine in 2-Month-Old Infants: A Double-Blind, Randomized Phase III Trial

**DOI:** 10.3390/vaccines10020254

**Published:** 2022-02-08

**Authors:** Yan Zheng, Zhifang Ying, Yanxiang Zou, Taotao Zhu, Dinggu Qian, Weixiao Han, Ya Jiang, Zhiwei Jiang, Xingyan Li, Jianfeng Wang, Jin Lei, Li Xu, Deyu Jiang, Changgui Li, Xiaoqiang Liu

**Affiliations:** 1Vaccine Clinical Research Center, Yunnan Provincial Center for Disease Control and Prevention, Kunming 650034, China; yaqueer_zy@163.com (Y.Z.); 18206869224@163.com (Y.Z.); 2Respiratory Virus Vaccine Department, Biological Product Institute, National Institutes for Food and Drug Control, Beijing 100050, China; yzf_0812@nifdc.org.cn (Z.Y.); changguili@aliyun.com (C.L.); 3Clinical Research Department, Sinovac Biotech Co., Ltd., Beijing 100089, China; zhutt@sinovac.com (T.Z.); hanwx@sinovac.com (W.H.); wangjianfeng@sinovac.com (J.W.); 4Vaccine Clinical Research Project Office, Qiubei County Center for Disease Control and Prevention, Wenshan Zhuang and Miao Autonomous Prefecture 663299, China; qblcmy@163.com; 5Vaccine Clinical Research Project Office, Mile County Center for Disease Control and Prevention, Honghe Hani and Yi Autonomous Prefecture 652399, China; mljkyfjz@163.com; 6Statistics Department, Beijing Key Tech Statistical Consulting Co., Ltd., Beijing 100025, China; zhi.wei.jiang@ktstat.com; 7Division of Acute Infectious Disease Control and Prevention, Yanshan County Center for Disease Control and Prevention, Wenshan Zhuang and Miao Autonomous Prefecture 663299, China; ys2345jk@163.com; 8Immunization Program Division, Gejiu County Center for Disease Control and Prevention, Honghe Hani and Yi Autonomous Prefecture 661099, China; ljin2000@126.com; 9Quality Assurance Department, Sinovac Biotech Co., Ltd., Beijing 100089, China; xul@sinovac.com; 10R&D Department, Sinovac Life Sciences Co., Ltd., Beijing 102629, China; jiangdy@sinovac.com

**Keywords:** Sabin strain, inactivated poliovirus vaccine, lot-to-lot consistency, safety, immunogenicity

## Abstract

Background: The Sabin-strain-based inactivated poliovirus vaccine (sIPV) plays an important role in poliomyelitis eradication in developing countries. As part of the phase III clinical development program, this study aimed to evaluate the safety, immunogenicity and lot-to-lot consistency of the sIPV in 2-month-old infants. Method: We conducted a phase III, randomized, double-blind, positive-controlled trial in which 1300 healthy infants were randomly assigned to four groups in a 1:1:1:1 ratio to receive one of the three lots of the sIPV or the control IPV at 2, 3 and 4 months of age. Serum samples were collected before the first dose and 30 days after the third dose of vaccination to assess the immunogenicity. Solicited local and systemic reactions were recorded within 7 days and unsolicited adverse events within 30 days after each vaccination. Results: Of the 1300 randomized infants, 1190 infants completed the study and were included in the per-protocol population. The seroconversion rates in the three lots of the sIPV were 95.67%, 97.03% and 95.59%, respectively, for type 1; 94.33%, 93.73% and 92.88%, respectively, for type 2; and 98.67%, 99.67% and 99.32%, respectively, for type 3. The ratios of GMTs for poliovirus types 1, 2 and 3 of each pair of lots were all between 0.67 and 1.50, therefore meeting the predefined immunological equivalence criteria. For the seroconversion rate of poliovirus types 1, 2 and 3, the pooled sIPV group was non-inferior to the IPV group. The incidence of solicited and unsolicited adverse reactions (ARs) was similar in the pooled sIPV lots and the IPV group, and most of them were mild to moderate in severity. Non-vaccine-related serious adverse events (SAEs) were reported. Conclusions: Three consecutive lots of sIPV demonstrated robust and consistent immunogenicity. The safety and tolerability of the sIPV was acceptable and similar to that of the IPV.

## 1. Introduction

Poliomyelitis (polio) is a serious infectious disease caused by the poliovirus, which mainly affects children under 5 years old [1]. Currently, there is no specific treatment for polio; the only way to prevent it is to be vaccinated. In 1988, the World Health Organization (WHO) launched the Global Polio Eradication Initiative (GPEI) and widely introduced polio vaccines worldwide, which induced a rapid decline in polio cases, from an estimated 350,000 cases in 1988 to 140 reported cases in 2020 [2,3]. China, the largest developing country in the world, has incorporated polio vaccines into its national routine program since 1982 and was declared polio-free in 2000 [4]. However, as two neighboring countries, Afghanistan and Pakistan, are still in a polio endemic, China is still highly vulnerable to the disease [5].

The oral poliovirus vaccine (OPV) and inactivated poliovirus vaccine (IPV) are two types of polio vaccines that have been widely implemented worldwide to eradicate poliovirus [6]. Although the efficacy and safety of the OPV has been demonstrated in recent decades, there are still potential risks of the OPV, including vaccine-associated paralytic poliomyelitis (VAPP), circulating vaccine-derived poliovirus (cVDPV) or immunodeficient vaccine-derived poliovirus (iVDPV) [7,8]. To end polio, including VAPP, cVDPV and iVDPV, as encouraged by the Polio Eradication and Endgame Strategic Plan 2013–2018 of GPEI [9], many high- and middle-income countries have switched their routine vaccination schedules from OPV-only to a sequential IPV-OPV or IPV-only for polio prevention [10]. In addition, the WHO decided on a globally synchronized switch from the trivalent OPV (tOPV) to bivalent type 1 and type 3 OPV (bOPV), with the aim of reducing polio cases with the type 2 OPV [11]. In most countries, the tOPV was successfully withdrawn from the immunization schedules in 2016, but it is still used in Afghanistan and Pakistan during special immunization activities. In China, a new polio immunization strategy (termed ‘‘2IPV + 2bOPV”) was implemented in October 2019, including Yunnan. This schedule involves the administration of the IPV at 2 months and 3 months of age, followed by the bOPV at four months and four years of age. Currently, the production of the IPV is mainly based on the Salk-strain and the Sabin-strain IPV. Considering the global shortage of the Salk-strain IPV and the high biosafety requirements and economic burden of the wild-type Salk-strain IPV production, replacing the wild-type Salk strain with the attenuated Sabin strain will have a lower biosafety risk and improve the availability and affordability of the IPV in low- and middle-income countries [12,13,14,15,16].

In China, Sinovac has developed a Sabin-based IPV (sIPV), which was prepared by a live-attenuated Sabin strain of poliovirus grown in Vero cells using microcarrier technology. The phase I and phase II randomized, controlled clinical trials have proven the safety and immunogenicity of the sIPV [17,18]. However, as part of the phase III vaccine clinical development program, the National Medical Products Administration (NMPA) typically expects to demonstrate a clinical assessment of manufacturing consistency by comparing the immunogenicity of three commercial batches of vaccines. Therefore, in this study, we aimed to evaluate the safety, immunogenicity and lot-to-lot consistency of the sIPV produced by Sinovac.

## 2. Materials and Methods

### 2.1. Study Design and Participants

The phase III, double-blind, randomized controlled trial was conducted in Mile city, Yanshan county and Qiubei county in Yunnan Province, China, from May 2020 to October 2020. Eligible participants were healthy infants aged 2 months (60–89 days) with proved legal identity. The exclusion criteria included the following: (1) axillary temperature > 37 °C; (2) prior receipt of poliomyelitis vaccine; (3) allergy or serious adverse reactions to any vaccine; (4) acute diseases within 7 days before vaccination; (5) any known immunodeficiency; (6) receipt of blood products within the previous 3 months; (7) receipt of any live-attenuated vaccines within the previous 14 days; (8) receipt of any subunit or inactivated vaccines within the previous 7 days; and (9) other conditions that were deemed not suitable for clinical trial.

After enrollment, all subjects were randomly assigned into four groups in a 1:1:1:1 ratio to receive three doses of either 3 lots of sIPV or IPV at 0, 1 and 2 months of age. Written informed consent was obtained from each infant’s parent or guardian before enrolment. The trial was conducted in accordance with Good Clinical Practice standards and the Declaration of Helsinki. This study was approved by the ethics committees of the Yunnan Provincial Center for Disease Control and Prevention (2019-5). This study was registered at ClinicalTrials.gov, accessed on 3 January 2022 (NCT04386707).

### 2.2. Vaccine

The investigational sIPV (0.5 mL/dose), developed by Sinovac Biotech Company, is a sterile, liquid trivalent vaccine for intramuscular injection. It was generated from Sabin poliovirus type 1, 2 and 3 strains grown on Vero cells. The antigen contents were 15, 45 and 45 D antigen units (DU) for type 1, 2 and 3 Sabin polioviruses, respectively, which were determined by the results of phase I and phase II clinical trials of the sIPV [16]. The control IPV (0.5 mL/dose) was produced by Sanofi Pasteur. Type 1 (Mahoney strain), 2 (MEF-1 strain) and 3 (Saukett strain) polioviruses grown on Vero cells were used to generate the control vaccine, with antigen contents of 40, 8 and 32 DU, respectively. The three consecutive commercial lots of sIPV vaccine (lot 201711001, expiry 22 November 2020; lot 201711002, expiry 29 November 2020; and lot 201711003, expiry 30 November 2020) and the single lot of IPV vaccine (lot P3M521M, expiry September 2020) were tested by the National Institutes for Food and Drug Control, China (NIFDC) and confirmed to adhere to the necessary specifications.

### 2.3. Randomization and Blinding

A random number table was generated by SAS 9.4 software (SAS Institute Inc., Cary, NC, USA) based on a preset block length to mask and label the study vaccines by independent biostatisticians. When enrolled in the study, every participant was assigned a unique number by order of entry and received the vaccine marked with the same number. All participants and investigators were blinded until the completion of the primary vaccination, a 30-day safety observation period and the measurement of neutralizing antibody.

### 2.4. Immunogenicity Assessment

Blood samples (3.0 mL) were collected from all participants on day 0 (before the first vaccination) and day 90 (30 days after the third vaccination).

After collection, the serum samples were separated, frozen and stored at the study site at −20 °C until assayed. The sera-neutralizing antibodies (NA) were detected in the NIFDC, and the detection method was microcytopathogenic effect assay [19]. The main immunogenic endpoints of the study were the seroconversion rate and the geometric mean titer (GMT) of NA. The seroconversion rate was defined as the percentage of participants with a reciprocal neutralizing antibody titer of either (1) < 1:8 before vaccination and ≥1:8 after vaccination or (2) ≥ 1:8 before vaccination and at least a 4-fold increase after vaccination. The immunogenicity assessment was conducted based on the per-protocol set (PPS), including infants who met eligibility criteria, complied with the protocol and produced immunogenicity results before and after vaccination.

### 2.5. Safety Assessment

Immediate adverse events (AEs) were observed on site in 30 min after each vaccination. A diary card was given to parents or guardians to record solicited local or systemic AEs occurring within 7 days, unsolicited AEs and serious adverse events (SAEs) occurring within 30 days after each dose. Solicited systemic AEs included fever, allergic reaction, skin and mucosa abnormality, irritability, decreased appetite, nausea/vomiting and diarrhea; solicited local AEs included pain, induration, redness, swelling, rash and pruritus. Any AEs were graded according to the guidelines issued by the NMPA [20]. The causal relationship between AEs and vaccination was judged by the investigators. Safety assessment was conducted based on the safety set (SS), which included infants who received at least one dose of a vaccine.

### 2.6. Statistical Analysis

The sample size was estimated using NCSS-PASS 11.0 software (NCSS, Kaysville, UT, USA). A total of 1300 participants (325 per group) were recruited, in order to allow for an approximate 20% drop-out rate. The sample size of each test group (sIPV lot) was 251, based on an equivalent design. The assessment indicator was GMT at the 30 days after immunization, and the equivalent threshold was between −0.176 and 0.176. The two-sided value of α was 0.05. The power of the overall test was 80%, and that of each test after adjustment was 97.78% (1–0.2/9 = 93.4%). Based on pre-clinical trial data, the estimated value of σ was 0.5. The sample size of the control group (IPV lot) was 251, as calculated according to a non-inferiority design. The assessment indicator was the seroconversion rate at 30 days after immunization. Based on our previous trial study, the estimated value of seroconversion rate was 90%. The ratio of the sample size of the test group to the control group was 3:1. The one-sided α level was 0.025. The non-inferior criterion was that the lower limit of 95% CI of the difference between the test group and the control group was greater than −10%.

Statistical analyses were performed using SAS version 9.4 (SAS Institute Inc., Cary, NC, USA). The immunologic equivalence of three consecutive lots of the sIPV was measured in terms of GMTs for each serotype included in the sIPV. Analysis of the log-transformed (base 10) titer was performed using an analysis of covariance (ANCOVA) model, with GMT pre-vaccination as the covariate and GMT post-vaccination as the dependent variable. The ratio of GMT after inverse logarithmic transformation in each of the two groups was calculated. Equivalence was demonstrated if the ratio of GMTs between each pair of lots was between 0.67 and 1.5. The two-sided 95% CI of the seroconversion rate was calculated to evaluate the difference between the pooled sIPV group and IPV group. If the lower limit was greater than −10%, the pooled sIPV group was considered to be non-inferior to the IPV group. A *p*-value of less than 0.05 was considered statistically significant.

## 3. Results

### 3.1. Study Population

The study process is shown in Figure 1. Between 11 May 2020 and 27 October 2020, a total of 1426 infants were screened, of whom 1300 were enrolled and randomly allocated to four groups in a 1:1:1:1 ratio. All participating infants received at least one dose of a vaccine and so were included in the safety set (SS). The number of subjects in the full analysis set (FAS) was the same as in the safety population. In total, 1190 (91.54%, 1190/1300) infants who completed the whole process of vaccination and blood sample collection before and after immunization within the time window were included in the per-protocol population for immunogenicity analysis. The demographic characteristics of the participants were similar in terms of age, ethnicity, gender, axillary temperature, height and weight across vaccine groups (Table 1).

### 3.2. Immunogenicity

The results for immunogenicity are shown in Table 2. Before vaccination, the GMTs and seropositive rates for type 1, type 2 and type 3 were balanced and comparable across the four groups. After vaccination, the seropositive rates for both type 1 and type 3 were 100% in all three sIPV lots, and for type 2, they were 100%, 100% and 99.66%. The seroconversion rate of the three lots of the sIPV group for type 1 were 95.67%, 97.03% and 95.59%; for type 2 they were 94.33%, 93.73% and 92.88%; and for type 3, they were 98.67%, 99.67% and 99.32%. The GMTs of the three lots of the sIPV group for type 1 were 3200.72, 2979.61 and 3265.02; for type 2 they were 570.92, 487.78 and 574.03; and for type 3, they were 1967.46, 1698.54 and 1978.96. The corresponding geometric mean increases (GMI) for type 1 were 216.45, 213.15 and 216.93; for type 2 they were 57.12, 48.92 and 50.35; and for type 3, they were 284.96, 262.62 and 282.44. For the IPV group, the seroconversion rates for type 1, type 2 and type 3 were 93.84%, 90.75% and 99.32%, respectively; GMTs for type 1, type 2 and type 3 were 577.98, 251.39 and 179.26, respectively; GMI for type 1, type 2 and type 3 were 39.93, 26.36 and 165.67, respectively.

### 3.3. Lot-to-Lot Consistency in sIPV Groups

Lot-to-lot equivalence was demonstrated for all three lots of sIPV for all three serotypes (Table 3). The ratios of the overall post-vaccination GMTs for type 1, 2 and 3 of each pair of lots were between 0.82 and 1.17, and the 95% CIs for these GMT ratios were all between 0.67 and 1.50.

### 3.4. Non-Inferiority between sIPV and IPV Group

After vaccination, the seroconversion rates in the pooled sIPV group and the IPV group were 96.10% and 93.84%, respectively, against type 1 poliovirus (*p* = 0.5910; difference: 2.27% (95% CI −0.14 to 5.81; 93.65% and 90.75%, respectively, against type 2 (*p* = 0.7664; difference: 2.90% (95% CI −0.44 to 7.03); and 99.22% and 99.32%, respectively, against type 3 (*p* = 0.3695; difference: −0.09% (95% CI −1.08 to 1.72) (Table 2). Therefore, the non-inferiority of the immunogenicity of the pooled sIPV group versus that of the IPV group for all three poliovirus serotypes was established.

### 3.5. Safety

The results of adverse reactions (ARs) are summarized in Table 4. The total AR rate was 40.38%, including 41.44% of the pooled sIPV group and 37.23% of the IPV group. There was no statistically significant difference in the AR rate between these two groups (*p* = 0.1920). No differences in the incidence of solicited and unsolicited ARs were observed between the pooled sIPV group and IPV group (*p* = 0.2934 and *p* = 0.4882). Most ARs were grade 1, and the total incidence of grade 1 ARs was 34.38% (pooled sIPV group vs. IPV group: 35.69% vs. 30.46%); 13.15% for grade 2 (13.23% vs. 12.92%); and 0.77% for grade 3 (0.82% vs. 0.62%). There was no grade 4 and above reported (Appendix A). The most common systemic reaction was fever; 19.62% of subjects had a fever (20.62% vs. 16.62%) followed by diarrhea, which occurred in 10.23% (10.26% vs. 10.15%) of subjects; there was no statistically significant difference of fever and diarrhea rates between these two groups (Appendix A). The overall incidence of serious adverse events (SAEs) was 6.23%, which was not related to vaccination. The incidence of SAEs in the pooled sIPV group and the IPV group was 6.26% and 6.15%, respectively, and the difference was not statistically significant (*p* = 1.0000).

The incidence of ARs in the three sIPV groups was 41.54%, 39.69% and 43.08%, and there was no statistically significant difference among groups (*p* = 0.6862). All of the local and systemic reactions had no significant difference among groups, except cough irritability (*p* = 0.0092) and inappetence (*p* = 0.0096) (Appendix A).

## 4. Discussion

In 2016, the Polio Eradication & Endgame Strategic Plan’s globally synchronized switch from the tOPV to bOPV was completed in all 155 OPV-using countries [10]. However, the ultimate goal of the world is completely to eradicate poliomyelitis. In this post-eradication era, the introduction of the IPV (even more than one dose) into the routine immunization programs of countries worldwide becomes increasingly important, and the demand for the IPV will also increase concomitantly. Given the strict biosafety requirements and high production costs of the conventional IPV, the use of a Sabin-based IPV is an affordable and practical option for polio eradication [14]. Additionally, the development of the Sabin IPV is also encouraged by the WHO.

This phase III, randomized, double-blind, controlled clinical trial aimed to evaluate the lot consistency, immunogenicity and safety of three lots of sIPV (Vero cell) in 2-month-old infants. The 95% confidence interval of the GMT ratio between each pair of sIPV lots all fell into the equivalence range, which proved that the three lots of sIPV reached the lot consistency criteria. This result indicates the stability and robustness of the vaccine production process. In addition, the non-inferiority of the seroconversion rate for the pooled sIPV lots compared to IPV at 30 days after vaccination was shown for all three poliovirus serotypes. This suggests that the investigational sIPV is as effective as the IPV in preventing poliovirus infection and can be used as a reliable substitute for conventional IPV. All investigational sIPV and control IPV in the current study exhibited good safety, and no vaccine-related SAEs were reported.

Our study showed that all three consecutive lots of sIPV were highly immunogenic against poliovirus types 1, 2 and 3. The seroconversion rates after vaccination in infants ranged from 95.67% to 97.03% for type 1 poliovirus, 92.88% to 94.33% for type 2 poliovirus and 98.67% to 99.67% for type 3 poliovirus. The finding was consistent with our previous phase III study [21], but these values are slightly lower than those of another double-blinded phase III clinical trial conducted in 2012-2014 [22], which may be due to the relatively high level of maternal antibodies in our study. Previous studies have shown that high levels of maternal poliovirus antibodies could attenuate the antibody responses to the IPVs [23,24,25]. Of note, in the present study, the post-vaccination GMTs of poliovirus types 1, 2 and 3 in the investigational sIPV group were noticeably higher than those in the IPV group. As the serum neutralizing test was conducted using the Sabin-strain virus, the sIPV antibodies had a stronger neutralizing capacity than the Salk-strain virus, which may explain the higher GMT in the investigational group in this study. In addition, the antigen contents between the sIPV and IPV groups were 15 vs. 40 DU/dose for type 1; 45 vs. 8 DU/dose for type 2; and 45 vs. 32 DU/dose for type 3. High D antigen content may lead to stronger immune responses and produce higher neutralizing antibody levels, but the immunogenicity of the three poliovirus types obtained from test sIPV were comparable to that of the control IPV. A possible explanation was that, in this study, the immunogenicity was evaluated by the seroconversion rate, and the non-inferiority test was carried out.

In accordance with the present findings, the three sIPV groups were comparable to the IPV group regarding safety, and no SAEs reported for the infants were associated with the vaccination. The overall incidence of ARs was between 39.69% and 43.08% in the three sIPV groups, and 37.23% in the IPV group. Most of the solicited local and systemic ARs caused by the sIPV and IPV were mild to moderate in severity. A small number of grade III ARs were observed, and the incidence of each group was less than 1%. The most common symptom of systemic solicited AR was fever, which occurred more frequently in the sIPV group than in the IPV group (20.62% vs. 16.62%). This finding is consistent with the results of previous studies, and a possible reason for this is that the D antigen in the investigational sIPV is higher than that in the control IPV [21,22,26]. A high D antigen content may induce adverse events, resulting in a higher frequency of fever in subjects inoculated with this dose [27]. Collectively, the symptoms of ARs in this study were basically consistent with previous studies. Thus, the studied vaccine was well tolerated.

There are several limitations to this study. First, the immunogenicity of the investigational sIPV might have been underestimated, as many infants have detectable maternal poliovirus antibodies before vaccination, and this may be due to the use of OPV on a large scale for many decades. Additionally, the decline of maternal antibodies over time was not taken into account. Another limitation was that our trial did not use the Salk strains from which the IPV was generated to measure the neutralizing antibodies induced by the two vaccines, which indicated that the finding that the increase in GMTs induced by sIPV is greater than that induced by IPV may be biased.

## 5. Conclusions

In summary, this study demonstrated that in 2-month-old infants, the sIPV elicited a consistent and robust immune response to all three poliovirus serotypes across three manufacturing scale lots. The safety and tolerability of the sIPV was acceptable and similar to the IPV. Thus, as a powerful vaccine for completely eradicating all types of poliovirus in the polio endgame period, the sIPV produced by Sinovac is stable and suitable for large-scale use.

## Figures and Tables

**Figure 1 vaccines-10-00254-f001:**
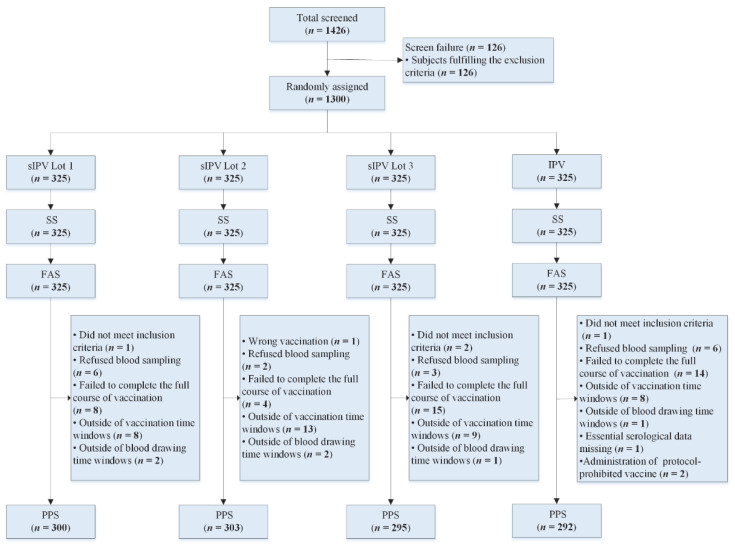
Trial profile. Flow chart of participants through screening, random assignment and analysis. Abbreviations: sIPV: Sabin-strain inactivated poliovirus vaccine (test group); IPV: inactivated poliovirus vaccine (control group); SS: safety set; FAS: full analysis set; PPS: per-protocol set.

**Table 1 vaccines-10-00254-t001:** Baseline characteristics and seroprevalence in per-protocol sets.

Characteristics	sIPV 1 (*n* = 300)	sIPV 2 (*n* = 303)	sIPV 3 (*n* = 295)	Pooled sIPV (*n* = 898)	IPV (*n* = 292)	*p* Value ^a^	*p* Value ^b^
Age (days) (mean ± SD)	71.4 ± 7.5	71.6 ± 8.1	70.9 ± 8.3	71.3 ± 8.0	71.3 ± 7.6	0.5351	0.9788
Han ethnic *n* (%)	96 (32.00)	108 (35.64)	94 (31.86)	298 (33.18)	97 (33.22)	0.2871	0.7365
Male *n* (%)	159 (53.00)	150 (49.50)	157 (53.22)	466 (51.89)	145 (49.66)	0.5923	0.5067
Axillary temperature (°C) (mean ± SD)	36.66 ± 0.26	36.67 ± 0.27	36.67 ± 0.27	36.67 ± 0.27	36.68 ± 0.27	0.7737	0.5027
Height (cm) (mean ± SD)	57.87 ± 2.34	57.79 ± 2.48	57.80 ± 2.54	57.82 ± 2.45	57.96 ± 2.57	0.9226	0.4147
Weight (kg) (mean ± SD)	5.72 ± 0.68	5.64 ± 0.70	5.70 ± 0.66	5.68 ± 0.68	5.71 ± 0.69	0.2802	0.6188
Poliovirus type 1							
Seropositivity rate (95%CI)	63.00 (57.26, 68.48)	62.05 (56.32, 67.53)	63.39 (57.61, 68.90)	62.81 (59.55, 65.98)	64.73 (58.95, 70.20)	0.9405	0.5544
GMT (95%CI)	14.79 (12.67, 17.26)	13.98 (12.03, 16.24)	15.05 (12.85, 17.63)	14.59 (13.35,15.95)	14.47 (12.46,16.81)	0.7842	0.9272
Poliovirus type 2							
Seropositivity rate (95%CI)	55.00 (49.18, 60.72)	53.47 (47.67, 59.19)	61.02 (55.19, 66.62)	56.46 (53.14, 59.73)	57.53 (51.64, 63.27)	0.1453	0.7473
GMT (95%CI)	10.00 (8.89, 11.24)	9.97 (8.89, 11.18)	11.40 (10.10, 12.87)	10.43 (9.74, 11.16)	9.54 (8.57, 10.62)	0.1972	0.1903
Poliovirus type 3							
Seropositivity rate (95%CI)	34.00 (28.65, 39.67)	31.35 (26.17, 36.91)	37.29 (31.75, 43.08)	34.19 (31.09, 37.39)	31.85 (26.54, 37.53)	0.3093	0.4626
GMT (95%CI)	6.90 (6.21, 7.67)	6.47 (5.87, 7.13)	7.01 (6.32, 7.77)	6.79 (6.40, 7.20)	6.51 (5.89, 7.20)	0.5071	0.4949

^a^ The *p* values were calculated for comparisons among 3 lots of sIPV group. ^b^ The *p* values were calculated for comparison of the pooled sIPV group and IPV group.

**Table 2 vaccines-10-00254-t002:** Seropositivity, seroconversion and GMTs in per-protocol sets 30 days after 3 doses.

Variable	sIPV 1 (*n* = 300)	sIPV 2 (*n* = 303)	sIPV 3 (*n* = 295)	Pooled sIPV (*n* = 898)	IPV (*n* = 292)	*p* Value ^a^	*p* Value ^b^	Difference (%(95% CI)) ^c^
Poliovirus type 1								
Seropositivity rate (95%CI)	100.00 (98.78, 100.00)	100.00 (98.79, 100.00)	100.00 (98.76, 100.00)	100.00 (99.59, 100.00)	100.00 (98.74, 100.00)	1.0000	1.0000	
Seroconversion rate (95%CI)	95.67 (92.70, 97.67)	97.03 (94.44, 98.63)	95.59 (92.58, 97.63)	96.10 (94.62, 97.27)	93.84 (90.43, 96.31)	0.5910	0.1029	2.27 (−0.41, 5.81)
GMT (95%CI)	3200.72 (2834.14, 3614.72)	2979.61 (2638.64, 3364.64)	3265.02 (2910.25, 3663.04)	3144.82 (2935.78, 3368.74)	577.98 (529.16, 631.31)	0.5329	<0.0001	
GMI (95%CI)	216.45(170.80, 274.29)	213.15(169.62, 267.86)	216.93(171.12, 275.00)	215.49 (188.34, 246.55)	39.93(33.47, 47.64)	0.9935	<0.0001	
Poliovirus type 2								
Seropositivity rate (95%CI)	100.00 (98.78, 100.00)	100.00 (98.79, 100.00)	99.66 (98.13, 99.99)	99.89 (99.38, 100.00)	100.00 (98.74, 100.00)	0.3285	1.0000	
Seroconversion rate (95%CI)	94.33 (91.08, 96.66)	93.73 (90.38, 96.18)	92.88 (89.32, 95.54)	93.65 (91.85, 95.16)	90.75 (86.83, 93.82)	0.7664	0.0929	2.90 (−0.44, 7.03)
GMT (95%CI)	570.92 (514.24, 633.84)	487.78 (440.81, 539.76)	574.03 (517.01, 637.34)	542.36 (510.92, 575.73)	251.39 (227.06, 278.33)	0.0450	<0.0001	
GMI (95%CI)	57.12 (47.64, 68.49)	48.92 (41.02, 58.35)	50.35 (41.99, 60.39)	52.01 (46.90, 57.68)	26.36 (22.18, 31.33)	0.4423	<0.0001	
Poliovirus type 3								
Seropositivity rate (95%CI)	100.00 (98.78, 100.00)	100.00 (98.79, 100.00)	100.00 (98.76, 100.00)	100.00 (99.59, 100.00)	100.00 (98.74, 100.00)	1.0000	1.0000	
Seroconversion rate (95%CI)	98.67 (96.62, 99.64)	99.67 (98.17, 99.99)	99.32 (97.57, 99.92)	99.22 (98.40, 99.69)	99.32 (97.55, 99.92)	0.3695	1.0000	−0.09 (−1.08,1.72)
GMT (95%CI)	1967.46 (1785.76, 2167.65)	1698.54 (1542.63, 1870.21)	1978.96 (1799.58, 2176.22)	1875.86 (1774.66, 1982.83)	1079.26 (986.16,1181.14)	0.0428	<0.0001	
GMI (95%CI)	284.96 (243.40, 333.62)	262.62(227.31, 303.41)	282.44 (243.50, 327.62)	276.41 (253.52, 301.36)	165.67 (144.46, 189.98)	0.7070	<0.0001	

^a^ The *p* values were calculated for comparisons among 3 lots of the sIPV group. ^b^ The *p* values were calculated for comparison of the pooled sIPV group and IPV group. ^c^ Non-inferiority was achieved, as the lower bound of the two-sided 95% CI was > −10%.

**Table 3 vaccines-10-00254-t003:** Equivalence between pairs of lots in the per-protocol set.

Serotype	Comparison	Adjusted GMT Ratio (95% CI)	Equivalence ^a^
Type 1	Lot 1 vs. Lot 2	1.10 (0.94, 1.27)	Yes
	Lot 1 vs. Lot 3	0.97 (0.84, 1.13)	Yes
	Lot 2 vs. Lot 3	0.89 (0.77, 1.03)	Yes
Type 2	Lot 1 vs. Lot 2	1.17 (1.02, 1.34)	Yes
	Lot 1 vs. Lot 3	0.96 (0.83, 1.10)	Yes
	Lot 2 vs. Lot 3	0.82 (0.71, 0.94)	Yes
Type 3	Lot 1 vs. Lot 2	1.17 (1.02, 1.34)	Yes
	Lot 1 vs. Lot 3	0.99 (0.87, 1.13)	Yes
	Lot 2 vs. Lot 3	0.85 (0.74, 0.97)	Yes

^a^ Lot-to-lot equivalence for each type was demonstrated if the two-sided 95% CI of the post-vaccination ratio of GMTs for that type in the two lots being compared was between 0.67 and 1.50.

**Table 4 vaccines-10-00254-t004:** Overall profiles of adverse reactions in the safety sets.

Adverse Reactions	sIPV 1 (*n* = 325)	sIPV 2 (*n* = 325)	sIPV 3 (*n* = 325)	Pooled sIPV (*n* = 975)	IPV (*n* = 325)	Total (*n* = 1300)	*p* Value ^a^	*p* Value ^b^
Overall	135 (41.54)	129 (39.69)	140 (43.08)	404 (41.44)	121 (37.23)	525 (40.38)	0.6862	0.1920
Solicited	130 (40.00)	125 (38.46)	132 (40.62)	387 (39.69)	118 (36.31)	505 (38.85)	0.8411	0.2934
Systemic	115 (35.38)	101 (31.08)	116 (35.69)	332 (34.05)	95 (29.23)	427 (32.85)	0.3867	0.1169
Local	27 (8.31)	39 (12.00)	25 (7.69)	91 (9.33)	30 (9.23)	121 (9.31)	0.1420	1.0000
Unsolicited	13 (4.00)	6 (1.85)	17 (5.23)	36 (3.69)	9 (2.77)	45 (3.46)	0.0625	0.4882

^a^ The *p* values were calculated for comparisons among 3 lots of the sIPV group. ^b^ The *p* values were calculated for comparison of the pooled sIPV group and IPV group.

## Data Availability

Not applicable.

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
