# Peer review of "Safety, Immunogenicity and Lot-to-Lot Consistency of Sabin-Strain Inactivated Poliovirus Vaccine in 2-Month-Old Infants: A Double-Blind, Randomized Phase III Trial"

_vaccines, 2022, doi:10.3390/vaccines10020254_

Round 1

Reviewer 1 Report

The WHO launched the Global Polio Eradication Initiative (GPEI) in 1988 with the aim of eradication poliomyelitis. The Polio Eradication and Endgame Strategic Plan includes a switch from live oral poliovirus vaccines (OPV) to inactivated poliovirus vaccines (IPV). Sabin IPV (sIPV) provide a means to lower biosafety risk of vaccine production while also increasing vaccine supplies for polio eradication. In this study, Zheng et. al. present findings from a phase III clinical trial to evaluate the safety, immunogenicity, and lot-to-lot consistency of sIPV in 2-month-old infants. The authors show similar levels of safety and immunogenicity of sIPV compared to Salk-based IPV. The authors also show consistent immunogenicity from 3 different lots of sIPV. This is a fairly straight forward manuscript that clearly presents data supporting the conclusions that sIPV is as safe and immunogenic as traditional IPV and that 3 lots of sIPV performed comparably.

The manuscript could be improved by addressing the comments listed below:

  1. The introduction should add text with regards to bOPV versus tOPV use as it relates to vaccination schedules (following what is provided in lines 73-74). Also, it is important to highlight that tOPV is still used in Afghanistan-Pakistan during special immunization activities. Also, additional text referencing IPV supply shortages would strengthen the authors argument for developing sIPV.
  2. The Figure 1 legend should include a description of what is shown in the trial profile as well as definitions for acronyms.
  3. The right side of Table 4 is cut-off.
  4. The authors should include text in the Discussion to elaborate on the different D antigen levels between IPV vs sIPV and how both vaccines achieved similar immunogenicity despite this. While the different D antigen levels are presented in the Methods, it would also be helpful to include this in the Discussion along with text shown in lines 297-301.
  5. The text, especially in the Introduction and Discussion needs to be revised for clarity. There are typos and awkward sentences scattered throughout the manuscript.

Author Response

Point 1: The introduction should add text with regards to bOPV versus tOPV use as it relates to vaccination schedules (following what is provided in lines 73-74). Also, it is important to highlight that tOPV is still used in Afghanistan-Pakistan during special immunization activities. Also, additional text referencing IPV supply shortages would strengthen the authors argument for developing sIPV.

 Response 1: Thanks for your constructive suggestions. In the introduction, we have added a description of the use of bOPV and tOPV in immunization programs, as well as the corresponding policy planning. At the same time, we highlighted the situation regarding the use of tOPV in Afghanistan-Pakistan. Finally, we added an argument about the current situation of IPV supply shortage and supplemented the corresponding reference [Ref. 13]. The relevant information has been shown on lines 75-79, 84.

Point 2: The Figure 1 legend should include a description of what is shown in the trial profile as well as definitions for acronyms.

Response 2: Thanks for your kind comments. We have added a legend to describe what is shown in the trial profile, and the definitions for acronyms. The relevant information has been shown on lines 208-2011.

Point 3: The right side of Table 4 is cut-off.

Response 3: Thanks for your kind comment. We have resized Table 4 to make it complete.

Point 4: The authors should include text in the Discussion to elaborate on the different D antigen levels between IPV vs sIPV and how both vaccines achieved similar immunogenicity despite this. While the different D antigen levels are presented in the Methods, it would also be helpful to include this in the Discussion along with text shown in lines 297-301.

Response 4: Thanks for your constructive suggestion, which will further improve our article. We have detailed the different D antigen contents between IPV and sIPV in the discussion section. High D antigen content may lead to stronger immune responses and yield a greater increase in antibody GMT. In our study, the seroconversion rate was used to evaluate immunogenicity. Although the seroconversion rate of sIPV group was slightly higher, the immunogenicity of the two groups is similar, because the non-inferiority criterion was that the lower limit of 95% CI of the difference between test group and control group was greater than -10%. The relevant information has been shown on lines 307-314.

Point 5: The text, especially in the Introduction and Discussion needs to be revised for clarity. There are typos and awkward sentences scattered throughout the manuscript.

Response 5: Thanks for your constructive suggestion. We have carefully checked the full manuscript and submitted it to the English Editing Services for high-quality editing. The relevant revision information has been shown on the manuscript.

Reviewer 2 Report

As indicated by the WHO, the cessation of OPV use is necessary to achieve polio eradication. Higher level of biological safety during the Sabin-IPV production respect to wild-IPV and evaluating the safety and immunogenicity of a candidate vaccine is crucial for Global Polio Eradication Initiative. Thanks to the authors for addressing this important topic.

1 You may consider more recent data (the number of worldwide polio cases has fallen from an estimated 350,000 in 1988 to 140 in 2020) available at the following link: https://www.cdc.gov/polio/what-we-do/

2 It may be helpful to add a brief description of the polio vaccine schedule in Yunnan province in the introduction section.

3 Line 94 There is Typo “;2”

4 Line 96 There is Typo “;;”

5 Lines 133 – Please add a reference about the method of micro cytopathogenic effect assay or describe it.

6 Lines 92-93 / Line 179 – The study period is not the same. Please revise it.

7 Line 218 There is Typo “for for”

Author Response

Point 1: You may consider more recent data (the number of worldwide polio cases has fallen from an estimated 350,000 in 1988 to 140 in 2020) available at the following link: https://www.cdc.gov/polio/what-we-do/.

Response 1: Thanks for your kind comment. We have replaced the data with more recent data according to your suggestion. The relevant information has been shown on line 61.

Point 2: It may be helpful to add a brief description of the polio vaccine schedule in Yunnan province in the introduction section.

Response 2: Thanks for your constructive suggestion. We have added a description of the polio vaccine schedule in Yunnan province in the introduction section. The relevant information has been shown on lines 79-82.

Point 3: Line 94 There is Typo “;2”

Response 3: Thanks for your suggestion. We have revised it and relevant information could be found on line 104.

Point 4: Line 96 There is Typo “;;”

Response 4: Thanks for your suggestion. We have revised it and relevant information could be found on line 107.

Point 5: Lines 133 – Please add a reference about the method of micro cytopathogenic effect assay or describe it.

Response 5: Thanks for your constructive suggestion. We have added a reference [Ref. 19] about the method of micro cytopathogenic effect assay in the methods section. The relevant information has been shown on lines 398-400.

Point 6: Lines 92-93 / Line 179 – The study period is not the same. Please revise it.

Response 6: Thanks for your kind comment. Our carelessness led to the error, and we have revised it. The relevant information has been shown on line 192.

Point 7: Line 218 There is Typo “for for”

Response 7: Thanks for your suggestion. We have revised it and relevant information could be found on line 235.

Reviewer 3 Report

The manuscript summarizes the phase III study of Sabin IPV performed in China. This reviewer has some comments and questions for it, and encourages the authors to consider them and rivise the manuscript.

Abstact, Line 37, "to receive one of the three lots of sIPV or the control IPV at 0, 1, and 2 months of age":  The authors collected infants aged 2 months, so they were immunized at 2, 3 and 4 months of age. They got 3 shots at intervals of one month. Is this correct? 

About the term of the phase III study, there is a conflict between the discription in the Materials and Methods (Line 92) and that in the Results (Line 179). Which is correct, 2019 or 2020?

About the vaccines used for the Phase III study – Three lots of Sabin IPV were used. This reviewer guesses from their lot numbering that they were produced in 2017. This reviewer thinks that the vaccines were used before they were expired. Can the authors indicate an expiration date? 

About Table 1 – This reviewer feels that the seropositivity rates were somewhat high although GMT values were not too high. The authors descibes in the Discussion that "many infants have detectable maternal po- 305 liovirus antibodies before vaccination, and this may be due to the use of OPV on a large scale for many decades" (Line 305). Is this discription the reason why some fractions of infants had  week immunity against poliviruses? If so, the authors should explainit in the main text.

Line 205; The author should write a full discription of "GMI".

Author Response

Point 1: Abstact, Line 37, "to receive one of the three lots of sIPV or the control IPV at 0, 1, and 2 months of age":  The authors collected infants aged 2 months, so they were immunized at 2, 3 and 4 months of age. They got 3 shots at intervals of one month. Is this correct?

Response 1: Thanks for your kind comment. Our carelessness led to the error, and we have revised it. The relevant information has been shown on line 37.

Point 2: About the term of the phase III study, there is a conflict between the discription in the Materials and Methods (Line 92) and that in the Results (Line 179). Which is correct, 2019 or 2020?

Response 2: Thanks for your constructive suggestion. 2020 is correct, and we have revised it in the Results.

Point 3: About the vaccines used for the Phase III study – Three lots of Sabin IPV were used. This reviewer guesses from their lot numbering that they were produced in 2017. This reviewer thinks that the vaccines were used before they were expired. Can the authors indicate an expiration date?

Response 3: Thanks for your suggestion. We have added the expiration date of each lot of vaccine in the vaccine part. The relevant information has been shown on lines 127-129.

Point 4: About Table 1 – This reviewer feels that the seropositivity rates were somewhat high although GMT values were not too high. The authors descibes in the Discussion that "many infants have detectable maternal poliovirus antibodies before vaccination, and this may be due to the use of OPV on a large scale for many decades" (Line 305). Is this discription the reason why some fractions of infants had weak immunity against poliviruses? If so, the authors should explain it in the main text.

Response 4: Thanks for your constructive suggestion. Infants with high levels of maternal poliovirus antibodies could attenuate the antibody responses to the IPVs and that’s why fractions of infants had weak immunity against poliovirus. And we have explained it in the discussion section, relevant information has been shown on lines 301-302.

Point 5: Line 205; The author should write a full discription of "GMI".

Response 5: Thanks for your suggestion. We have added the full description “geometric mean increase” of “GMI” on line 222.